# DLm6Am: A Deep-Learning-Based Tool for Identifying N6,2′-O-Dimethyladenosine Sites in RNA Sequences

**DOI:** 10.3390/ijms231911026

**Published:** 2022-09-20

**Authors:** Zhengtao Luo, Wei Su, Liliang Lou, Wangren Qiu, Xuan Xiao, Zhaochun Xu

**Affiliations:** 1Computer Department, Jingdezhen Ceramic University, Jingdezhen 333403, China; 2School of Life Science and Technology and Center for Informational Biology, University of Electronic Science and Technology of China, Chengdu 610054, China

**Keywords:** N6,2′-O-dimethyladenosine, m^6^Am site identification, deep learning

## Abstract

N6,2′-O-dimethyladenosine (m^6^Am) is a post-transcriptional modification that may be associated with regulatory roles in the control of cellular functions. Therefore, it is crucial to accurately identify transcriptome-wide m^6^Am sites to understand underlying m^6^Am-dependent mRNA regulation mechanisms and biological functions. Here, we used three sequence-based feature-encoding schemes, including one-hot, nucleotide chemical property (NCP), and nucleotide density (ND), to represent RNA sequence samples. Additionally, we proposed an ensemble deep learning framework, named DLm6Am, to identify m^6^Am sites. DLm6Am consists of three similar base classifiers, each of which contains a multi-head attention module, an embedding module with two parallel deep learning sub-modules, a convolutional neural network (CNN) and a Bi-directional long short-term memory (BiLSTM), and a prediction module. To demonstrate the superior performance of our model’s architecture, we compared multiple model frameworks with our method by analyzing the training data and independent testing data. Additionally, we compared our model with the existing state-of-the-art computational methods, m6AmPred and MultiRM. The accuracy (ACC) for the DLm6Am model was improved by 6.45% and 8.42% compared to that of m6AmPred and MultiRM on independent testing data, respectively, while the area under receiver operating characteristic curve (AUROC) for the DLm6Am model was increased by 4.28% and 5.75%, respectively. All the results indicate that DLm6Am achieved the best prediction performance in terms of ACC, Matthews correlation coefficient (MCC), AUROC, and the area under precision and recall curves (AUPR). To further assess the generalization performance of our proposed model, we implemented chromosome-level leave-out cross-validation, and found that the obtained AUROC values were greater than 0.83, indicating that our proposed method is robust and can accurately predict m^6^Am sites.

## 1. Introduction

More than 160 RNA modification types have been discovered so far [1]. Among them, N6-methyladenosine (m^6^A) is the most widespread post-transcriptional modification of lncRNA and mRNA in mammalian cells [2]. There is another reversible modification termed N6,2′-O-dimethyladenosine (m^6^Am), which was originally found at the 5′ end of mRNA in viruses and animal cells in 1975 [3]. Different from m^6^A modification, m^6^Am is a terminal modification, which is usually 2′-O-methylated at the second base adjacent to the 5′ cap in many mRNAs and is further methylated at the N^6^ position [4] (see Figure 1). The RNA modification m^6^Am is catalyzed by phosphorylated C-terminal domain (CTD)-interacting factor 1, i.e., PCIF1 [5], a writer protein specific to the cap-related m^6^Am, and could be demethylated by the fat mass and obesity-associated protein, i.e., FTO [6], one of the m^6^A demethylases. Thus, m^6^Am is regulated by PCIF1 and FTO dynamically, educing the direction of the cap epitranscriptomics.

Since the initial discovery of m^6^Am modification, some scholars have started to reveal its function. Several recent studies demonstrated that m^6^Am might be associated with higher protein expression [4], obesity-related translation regulation [4], increased translation efficiency [7,8], and mRNA stability [6,9,10]. Mauer et al. [6] found that the stability of transcripts that begin with m^6^Am was enhanced, while the stability of m^6^Am mRNAs could be reduced once demethylated by FTO. However, this observation was recently challenged. Wei et al. [11] reported that the expression levels of transcripts possessing cap m^6^Am seemed not to change with the knockdown of FTO. Moreover, another two studies [5,8] suggested that the loss of m^6^Am in PCIF1 knockout (KO) cells did not markedly affect the level of transcripts with m^6^Am. More corroborating evidence is needed to support the conclusion that m^6^Am can influence mRNA stability. On the other hand, it seems that m^6^Am modification can also affect mRNA translation. The positive association between m^6^Am methylation level and translation level was revealed using ribosome profiling experiments [8]. However, the biological function of m^6^Am has largely remained a mystery due to the lack of robust methods for sensitively identifying this modification at the transcriptome-wide level. Consequently, accurate identification of transcriptome-wide m^6^Am sites is crucial to understanding and exploring underlying m^6^Am-dependent mRNA regulation mechanisms and biological functions.

There have been efforts aiming to identify m^6^Am sites with wet-lab experimental methods [12,13,14]. However, such experimental methods are still expensive and time-consuming; thus, the development of computational approaches to accurately identify m^6^Am sites is urgently needed. Recently, researchers have attempted to computationally identify m^6^Am sites with machine-learning algorithms. In this field, Meng’s team first developed a computation method named m6AmPred [15] based on sequence-derived information in 2021. They collected m^6^Am sequencing data generated by miCLIP-seq technology from [6,16], selected 41 nt sequences centered on Adenosine (A) as positive samples, and randomly chose 41 nt sequences centered on A in non-modified BCA (B = C, G, or U) motif as negative samples. Electron–ion interaction potential and pseudo-EIIP (EIIP-PseEIIP) were employed to encode each mRNA sequence, and the eXtreme Gradient Boosting with Dart algorithm (XgbDart) was used to build the m^6^Am site predictor. In the same year, this team collected experimental data of twelve widely occurring RNA modifications, including m^6^Am sequencing data generated by miCLIP-seq [6,10,16]. Additionally, they developed an attention-based multi-label deep learning framework named MultiRM [17], consisting of an embedding module and an LSTM-Attention block, which can simultaneously predict the putative sites of twelve RNA modifications covering m^6^Am.

Despite recent advances in the computational identification of m^6^Am sites, some limitations and shortcomings still exist, as shown below. (i) Current computational methods trained on m^6^Am data from miCLIP-seq [6,10,16] are limited by the confidence level of training data. The experimental method miCLIP-seq for mapping m^6^Am relies on m^6^A antibodies, which is not good enough to distinguish m^6^Am from 5′-UTR m^6^A. Such an indirect approach is constrained by the limited activity of 5′ exonuclease, inaccuracy of TSS annotation, and low efficiency of UV crosslinking, thereby greatly reducing the precision and sensitivity of m^6^Am identification. Moreover, data sources [6,10,16] of current methods only provide m^6^Am peak regions with lengths ranging from 100 to 250 nt rather than the m^6^Am site at single-nucleotide resolution. More high-confidence data of m^6^Am sites are needed to construct computational models for m^6^Am site identification. (ii) Current computational methods do not implement sequence-redundancy removal. Generally, to reduce the redundancy in sample sequences, the CD-HIT-EST tool [19] is employed to remove sequences with sequence similarity greater than a certain threshold (typically 80%). After sequence-redundancy removal with the threshold set as 80%, we found that the original positive and negative datasets provided by m6AmPred lost 55 and 3630 sequences, respectively. (iii) The generalization performance of current computational methods remains to be further validated.

Recently, a sensitive and specific approach termed m^6^Am-seq [14] has been developed to directly profile transcriptome-wide m^6^Am, which can provide high-confidence m^6^Am site data at the single-nucleotide level to develop computational methods of m^6^Am site detection and future functional studies of m^6^Am modification. Moreover, in the field of m^6^A site prediction, MultiRM [17], DeepM6ASeq [20], and MASS [21] were successfully developed using an effective hybrid framework embedding with CNN and LSTM and achieved promising performance. Inspired by these single-nucleotide m^6^Am data and successful applications of deep learning frameworks, here, we present DLm6Am, an attention-based ensemble deep-learning framework to accurately identify m^6^Am sites. We firstly discuss the effect of several frequently used RNA sequence encoding methods, including one-hot, nucleotide chemical property (NCP), and nucleotide density (ND), on six different classifiers, including random forest (RF), support vector machine (SVM), eXtreme Gradient Boosting (XGBoost), Bi-directional long short-term memory (BiLSTM), convolutional neural network (CNN), and the embedding model CNN-BiLSTM. We found that the embedding deep learning model CNN-BiLSTM trained by fusion features could achieve the best prediction performance. Subsequently, we compared the prediction performance of the embedding deep learning model with and without an attention layer. Next, after ranking ACC values of embedding deep learning models with an attention layer under different hyper-parameters, we selected three models with the top three ACC values as base classifiers and adopted a voting strategy to build the final ensemble deep-learning model. To further assess the generalization performance of our proposed model, we implemented an independent test and also employed chromosome-level leave-one-out validation. Finally, a user-friendly webserver was established based on the proposed model and made freely accessible at http://47.94.248.117/DLm6Am/ (accessed on 15 September 2022) [22] to serve the research community. Moreover, the source code was provided in a Github repository (https://github.com/pythonLzt/DLm6Am (accessed on 15 September 2022)) to enable future uses.

## 2. Results and Discussion

### 2.1. Overview of DLm6Am

DLm6Am can identify m^6^Am modification sites from RNA sequences by the following steps (see Figure 2): (1) extracting m^6^Am sequences from the data generated by the sensitive and specific approach termed m6Am-seq, and constructing positive and negative samples using CD-HIT-EST tool; (2) generating context-based features using one-hot, NCP, and ND encoding schemes based on the primary RNA sequences; (3) establishing the ensemble-leaning-based classification model based on three CNN-BiLSTM-attention models with the top three ACC values; (4) averaging output from all the individual base classifiers as the final decision to achieve m^6^Am or non-m^6^Am prediction.

Specifically, although three base classifiers have different hyper-parameters, they possess similar architecture, mainly consisting of three parts (see Figure 2). The first module is a multi-head attention layer, which can capture the importance feature scores from individual input positions along the input sequence, thereby enhancing the learning ability of the prediction model. The second module is an embedding module that took the output of the multi-head attention layer as input and fed them into two parallel deep learning models, a CNN and a BiLSTM, respectively. The sub-module CNN, which could extract the hidden contextual features of RNA sequences, included two convolution layers to extract different input features and a maximum pooling layer to reduce the dimension of the extracted features and the amount of calculation. Moreover, the sub-module BiLSTM could capture possible long-range dependencies of the features. A fully connected layer (FCN) was followed by BiLSTM for feature dimension reduction, and the rectified linear unit (ReLU) [23] was used as the activation function to improve the computational efficiency and retain the gradient. After, we combined the two kinds of features extracted from the embedding module and fed them into the third module, a prediction module, which consists of two fully connected layers. Each of these two fully connected layers is succeeded by a dropout operation to mitigate overfitting. The Softmax function [24] was applied on the last layer to predict whether the central nucleotide A of the given RNA sequence is m^6^Am or non-m^6^Am site. More details are given in Appendix A regarding the actual model configurations used, such as layer sizes, depth, and the number of parameters.

### 2.2. Comparison with Different Model Architectures

To demonstrate the superior performance of our model architecture, we compared multiple model frameworks with our method by analyzing the training data described in Section 3.1, “Benchmark dataset”, using the fusion features generated by binary encoding, NCP, and ND. The competitors mainly contained classical traditional classifiers, such as random forest (RF), support vector machine (SVM), and eXtreme Gradient Boosting (XGBoost); and deep learning feature extractors, such as CNN, BiLSTM, CNN-BiLSTM, and CNN-BiLSTM-attention. Among them, CNN-BiLSTM represents the embedding model with CNN and BiLSTM, while CNN-BiLSTM-attention represents the embedding model generated by CNN with an attention layer and BiLSTM with an attention layer. As mentioned above, our model DLm6Am is an ensemble deep learning framework integrated by three CNN-BiLSTM-attention models with the top three ACC values.

To demonstrate the stability of models, we implemented five-fold cross-validation 20 times for each model with tuned hyper-parameters. The detailed configuration of models is found in Appendix A. All detailed results generated by five-fold cross-validation on training data, including the area under the PR (Precision and Recall) curves and ROC curves, are found in Table 1. It can be seen that the standard deviations for most metrics of models are small, demonstrating the good fitness and stability of these models. Additionally, as shown in Table 1, it can be concluded that the average values of important indicators of deep-learning models were higher than those of traditional classifiers. For example, compared to the traditional classifier RF, the ACC, MCC, AUROC, and AUPR average values of the CNN model were higher by 1.02%, 2.12%, 2%, and 3.22%, respectively, while the ACC, MCC, AUROC, and AUPR average values of the BiLSTM model were increased by 0.34%, 0.71%, 1.08%, and 2.29%, respectively. Especially, our model DLm6Am was improved by 3.25%, 6.5%, 2.36%, and 3.45% in terms of ACC, MCC, AUROC, and AUPR average values, respectively, compared to RF.

Additionally, we implemented a series of ablation tests to demonstrate the superiority of our model architecture. Firstly, the framework without ensemble learning, i.e., CNN–BiLSTM–attention was included in comparison with our proposed model, DLm6Am. It can be seen from Table 1 that the ACC, MCC, AUROC, and AUPR average values of the CNN–BiLSTM–attention were reduced by 0.35%, 0.7%, 0.21%, and 0.44%, respectively. This demonstrates that the prediction performance for m^6^Am site identification can be further improved using ensemble learning. Secondly, the embedding architecture without the attention module CNN-BilSTM was also compared with DLm6Am and CNN–BiLSTM–attention. The CNN–BiLSTM–attention framework achieved higher performance than the embedding architecture without an attention layer in terms of ACC, MCC, AUROC, and AUPR. This indicates that the attention mechanism has the capacity to identify m^6^Am key information, thereby improving model performance. The metric values of DLm6Am were improved overall compared to those of CNN-BilSTM, indicating that ensemble learning and the attention mechanism are of crucial importance in identifying m^6^Am sites. Moreover, the embedding deep learning model CNN-BiLSTM is superior to single deep learning models in terms of ACC and MCC, because the embedding deep learning model can gain the advantage of CNN and BiLSTM to simultaneously capture possible local-range and long-range dependencies of the features. Compared with single deep learning models, the performance of our model has obvious advantages, e.g., our model DLm6Am was improved by 2.77%, 3.04%, 2.91%, 5.79%, 1.28%, and 1.16% over BiLSTM in terms of average values of Sn, Sp, ACC, MCC, AUROC, and AUPR, respectively.

As some of the differences are minimal when compared with other methods, on the one hand, we sought to find an effective solution for the additional performance gains of the DLm6Am model while varying model complexity (e.g., the number of layers/nodes), as well varying/increasing training set size. The results listed in Appendix A show that DLm6Am can achieve better prediction performance under the current model configurations used in this study. On the other hand, we calculated the statistical significance of the observed differences using the Wilcoxon test. A graphical representation of results, shown in Figure 3, was mainly created by the R package “ggplot2”, which can provide a single layer geom_signif to calculate the significance of a difference between metric values of DLm6Am and other models. The level of significance (NS, *, **, ***) was added to the plot in a single line. As seen, with the exception of AUPR values of DLm6Am and CNN, the observed differences between DLm6Am and other models are significant. Based on these results, the embedding architecture CNN–BiLSTM–attention and ensemble learning were chosen to build the DLm6Am model.

To further confirm the superiority of our proposed framework, we compared the prediction performance of DLm6Am and other combinations of different base classifiers, such as SVM, RF, XGBoost, LSTM, and CNN. The ensemble classification results are listed in Appendix A. It can be seen that the DLm6Am achieved better prediction performance than other ensemble combinations.

Next, we used the independent testing data mentioned in Section 2.1, “Benchmark dataset”, to evaluate the generalization performance of these different models. All the testing results are listed in Table 2. Additionally, Figure 4 illustrates the comparison results based on PR curves and the ROC curves of different methods. It can be seen from Table 2 and Figure 4 that our proposed DLm6Am are generally better than the other models, further demonstrating the advantages of embedding deep-learning architecture and ensemble learning. These independent testing results indicate that our method can be used to accurately identify novel m^6^Am sites that have not been seen in training data.

### 2.3. Hold-Out Cross-Validation on Chromosome Level

Hold-out cross-validation is generally an effective method to validate the generalization performance of the proposed models. In this study, for chromosomes associated with data greater than 50, we used data of each chromosome as the testing data to evaluate the performance of the model trained on the data of the remaining chromosomes. Additionally, because the data sizes of chromosomes 13, 18, 20, 21, and 22 are all less than 50, these data were integrated as one testing dataset, while the remaining data of chromosomes were used as a training dataset. For all hold-out cross-validation, models were fitted on the training data using the same hyper-parameters of DLm6Am for the strict evaluation of generalization performance.

The results using hold-out cross-validation on chromosome level are reported in Table 3. It can be clearly seen that the most important metric, AUROC, is greater than 0.83, indicating that our proposed method is robust and can accurately predict m^6^Am sites on different chromosome data.

### 2.4. Comparison with Existing Methods

To further assess the performance of DLm6Am, we compared our model with the other existing state-of-the-art computational methods, MultiRM and m6AmPred, to identify m^6^Am sites in RNA sequences. Among them, MultiRM consists of three parts, in which the first module takes the one-hot encoding of RNA sequences as input and adopts three different embedding techniques to embed features; then, the embedding features are fed into an LSTM and an attention layer. For m6AmPred, electron–ion interaction potential (EIIP) and Pseudo-EIIP (PseEIIP) were used as encoding schemes to represent the sample sequences, and the eXtreme Gradient Boosting with Dart algorithm (XgbDart) was employed to build the final model. Here, to provide a fair performance comparison, we applied an identical encoding scheme and classification algorithm to the training data used in this study and used independent testing data to evaluate the constructed model.

All the results using five-fold cross-validation on training data and using independent testing data are deposited in Figure 5a,d, respectively. It can be seen clearly that DLm6Am showed better predictive performance than the other two predictors. More specifically, Sn, Sp, ACC, MCC, AUROC, and AUPR for the DLm6Am model outperformed m6AmPred using five-fold cross-validation on training data, by 5.43%, 6.84%, 6.13%, 12.26%, 4.60%, and 4.27%, respectively. Sn, Sp, ACC, MCC, AUROC, and AUPR of DLm6Am were higher than those of MultiRM by 5.43%, 11.42%, 8.42%, 17.14%, 7.18%, and 7.55%, respectively. Moreover, in comparison with m6AmPred using independent testing data, Sn, Sp, ACC, MCC, AUROC, and AUPR for the DLm6Am model were improved by 9.60%, 3.32%, 6.45%, 12.95%, 4.28%, and 4.26%, respectively. Compared with MultiRM, these metric values were increased by 3.12%, 13.74%, 8.42%, 16.43%, 5.75%, and 6.57%, respectively. This result suggests that joint use of the hybrid architecture CNN–BiLSTM–attention and ensemble learning has a strong potential for application in other modification site prediction tasks in RNAs.

Additionally, both ROC and PR curves were plotted to demonstrate the performance of MultiRM, m6AmPred, and DLm6Am (Figure 5). The DLm6Am model had much higher performance than MultiRM and m6AmPred, further illustrating the stability and generalization ability of our proposed model, DLm6Am.

### 2.5. Webserver Functionality

To achieve quick prediction of m^6^Am sites from RNA sequences, a user-friendly web interface DLm6Am was developed based on python using the web micro-framework Flask. The webserver DLm6Am has several user interfaces and provides multiple functions, including m^6^Am site prediction, the introduction of this webserver, data download, and citation of the relevant literature. DLm6Am allows users to perform prediction by typing or copying/pasting the query RNA sequences with FASTA format into the input box. After a short online wait, the results will be displayed on the webserver. Additionally, the users can receive the prediction results by email without a long wait after uploading a FASTA file with multiple RNA sequences of interest and inputting the email address and job IDs. In summary, this service allows researchers to identify the transcriptome-wide m^6^Am sites, thereby enabling researchers to understand underlying m6Am-dependent mRNA regulation mechanisms and biological functions.

## 3. Materials and Methods

### 3.1. Benchmark Dataset

Recently, Sun et al. [14] developed a sensitive and direct method named m^6^Am-seq to identify transcriptome-wide m^6^Am and provided 2166 high-confidence m^6^Am sites at single-nucleotide resolution throughout the human transcriptome. In terms of these site information and the human genome assembly hg19, we can extract corresponding sample sequences using the (2*ξ* + 1)-nt long sliding window, formulated as below.
(1)$$R_{\xi }\left(A\right)=R_{-\xi }R_{-\left(\xi -1\right)}\cdots R_{-2}R_{-1}AR_{+1}R_{+2}\cdots R_{+\left(\xi -1\right)}R_{+\xi }$$
where the double-line character A represents the nucleotide adenosine in BCA (B = C, G, or U) motifs, the value of subscript ξ is an integer, $R_{-\xi }$ represents the ξ-th upstream nucleotide from the center, $R_{+\xi }$ represents the ξ-th downstream nucleotide from the center, and so on. In this study, after preliminary analysis, *ξ* value was set to 41. If the centers of RNA sequence segments are the experimentally confirmed m^6^Am sites, these sequences are regarded as positive candidate samples and are put into the positive dataset $S^{+}$. Otherwise, the RNA sequence segments are considered negative samples and are classified into the negative dataset $S^{-}$. After reducing the sequence identity to 80% by using CD-HIT-EST tool [19], we randomly selected negative samples at 1:1 positive-to-negative ratio to construct the final benchmark dataset S ($S=S^{+}\cup S^{-}$), summarized in Table 4. For constructing and training the prediction model, we randomly sampled 80% of data from the benchmark dataset as training set, and the remaining 20% was considered as independent testing dataset to test the prediction performance of the constructed model. This ratio of the training/testing set was obtained by analyzing the effect of training set size on performance (Appendix A). The training and independent testing can be downloaded from http://47.94.248.117/DLm6Am/download (accessed on 15 September 2022) [22].

### 3.2. Feature Extraction from RNA Sequence

It is essential to carefully design and extract features to develop robust and reliable computational methods of m^6^Am site identification. Various feature-encoding methods have been developed to represent and convert RNA sequences into numeric vectors, such as context-based features, structure-based features, and integrated features. Majority of these features could be easily generated by useful bioinformatics tools, such as PseKNC [25], iLearn [26], and iLearnPlus [27]. In this study, we selected the most prevalent context-based features to represent RNA sequence fragments, including one-hot, nucleotide chemical property (NCP), and nucleotide density (ND).

#### 3.2.1. Binary Encoding of Nucleotide

Binary encoding of nucleotide is a simple and efficient method to characterize sample sequences of protein, DNA, or RNA. Generally, four types of nucleotide, i.e., A (adenine), C (cytosine), G (guanine), and U (uracil), can be encoded by (1, 0, 0, 0), (0, 1, 0, 0), (0, 0, 1, 0), and (0, 0, 0, 1), respectively. Thus, each sample sequence with 41 nt length can be transformed into a 41 × 4 numerical matrix.

#### 3.2.2. Nucleotide Chemical Property (NCP)

In terms of ring structures (two for purines, i.e., A and G, one for pyrimidines, i.e., C and U), chemical functionality (amino group for A and C, keto group for G and U), and strength of hydrogen bonding interaction between the base pairs (stronger between C and G, weaker between A and U), the four types of nucleotides can be classified into three different categories. Thus, the i-th nucleotide of a given sample sequence formulated by Equation (1) can be represented by a 3-dimensional vector, as shown below.
(2)$$R_{i}=\left(x_{i},\;y_{i},\;z_{i}\right)$$
where $x_{i}=\left\{\begin{array}{rr} \begin{array}{rr} 1 & 0 \end{array} & \begin{array}{rr} R_{i}\in \left\{A,G\right\} & R_{i}\in \left\{C,U\right\} \end{array} \end{array}\right.$ represents ring structures, $y_{i}=\left\{\begin{array}{rr} \begin{array}{rr} 1 & 0 \end{array} & \begin{array}{rr} R_{i}\in \left\{A,C\right\} & R_{i}\in \left\{G,U\right\} \end{array} \end{array}\right.$ for chemical functionality, and $z_{i}=\left\{\begin{array}{rr} \begin{array}{rr} 1 & 0 \end{array} & \begin{array}{rr} R_{i}\in \left\{A,U\right\} & R_{i}\in \left\{C,G\right\} \end{array} \end{array}\right.$ for hydrogen bond. Therefore, each sample sequence with 41 nt length can be converted into a 41 × 3 numerical matrix.

#### 3.2.3. Nucleotide Density (ND)

Nucleotide density, also termed accumulated nucleotide frequency (ANF), represents the accumulated frequency distribution of the nucleotide at each position along a given sample sequence. Specifically, nucleotide density of the nucleotide $R_{i}$ in the sample sequence $R_{\xi }\left(A\right)$ can be calculated by the number of occurrences of the nucleotide $R_{i}$ in the region from position 1 to position i divided by the length i of this region. For example, if the given sequence is “ACGACUUAGA”, it can be transformed into a numerical vector (1, 1/2, 1/3, 2/4, 2/5, 1/6, 2/7, 3/8, 2/9, 4/10). Generally, this feature can be combined with NCP to form a widely applied fusion feature named NCPD [28,29,30,31,32,33,34,35,36].

### 3.3. Classification Method

Generally, three types of major prediction algorithms have been employed to implement various prediction tasks in bioinformatics field, including (i) scoring-function-based methods, such as positional weight matrix (PWM) [37]; position-correlation scoring function (PCSF) [38] and relative stability (DE) [39]; (ii) traditional machine-learning-based method, such as random forest (RF) [40], support vector machine (SVM) [41], and decision tree (DT) [42]; and (iii) deep-learning-based methods, such as convolutional neural network (CNN) [43] and long short-term memory (LSTM) [44].

Deep-learning-based methods have been widely applied in biological research [17,45,46,47,48,49], especially for CNN and LSTM. The CNN framework can depict latent information of sequential features by integrating local dependencies, and LSTM architecture can capture possible long-range dependencies of the features. Recently, an effective hybrid framework embedding with CNN and LSTM has been successfully used in m^6^A site prediction, such as MultiRM [17], DeepM6ASeq [20], and MASS [21]. Inspired by these successful applications, in this study, we constructed an embedded deep-learning model with multi-head attention layer using CNN and BiLSTM to identify m^6^Am sites from RNA sequences. Finally, to further improve the generalization performance, we selected three embedded models with top three ACC values under different hyper-parameter combinations as base classifiers and adopted voting strategy to obtain the final ensemble deep-learning model, named DLm6Am. To demonstrate the superiority of our model architecture, we designed a series of ablation tests, including the model architecture without ensemble learning (CNN-BiLSTM-attention), without the attention layer (CNN-BiLSTM), and single deep-learning models.

### 3.4. Performance Evaluation

Prediction performance of the proposed models is generally measured using several metrics, such as sensitivity (*Sn*), specificity (*Sp*), accuracy (*ACC*), and Matthews correlation coefficient (*MCC*), formulated by the following equation.
(3)$$\left\{\begin{array}{c} Sn=\frac{TP}{TP+FN}\;\\ Sp=\frac{TN}{TN+FP}\;\\ \begin{array}{c} Acc=\frac{TP+TN}{TP+TN+FP+FN}\;\\ MCC=\frac{TP\times TN-FP\times FN}{\sqrt{\left(TP+FN\right)\times \left(TN+FN\right)\times \left(TP+FP\right)\times \left(TN+FP\right)}} \end{array} \end{array}\right.$$
where *TP*, *FN*, *TN*, and *FP* stand for the numbers of true positives, false negatives, true negatives, and false positives, respectively. Additionally, the area under receiver operating characteristic curve (AUROC) and the area under precision and recall curves (AUPR) are usually calculated to evaluate the prediction performance. If their value is higher, the prediction performance is better.

## 4. Conclusions

In this study, we have developed a new computational method, called DLm6Am, to predict the transcriptome-wide likelihoods of m^6^Am sites. DLm6Am is a deep-learning-based framework integrated by three base classifiers with the top three ACC values, which have similar network architectures. Each of the base classifiers has three parts, a built-in attention module to help extract useful sequence patterns, an embedding module consisting of a CNN and a BiLSTM for extracting features from RNA sample sequences, and a prediction module consisting of two fully connected layers to obtain a final prediction decision. The ablation tests in Section 3.2 showed the superiority of our designed framework. Moreover, independent tests and hold-out cross-validation on the chromosome level have demonstrated the generalization capacity of our model in predicting novel m^6^Am sites. Comparison results on benchmark datasets have illustrated the superior prediction performance of DLm6Am over other state-of-the-art methods, m6AmPred and MultiRM. However, currently, we only used limited experimental data to construct the model for m^6^Am site prediction. In the future, we will collect large-scale experimental data to develop better prediction algorithms.

## Figures and Tables

**Figure 1 ijms-23-11026-f001:**
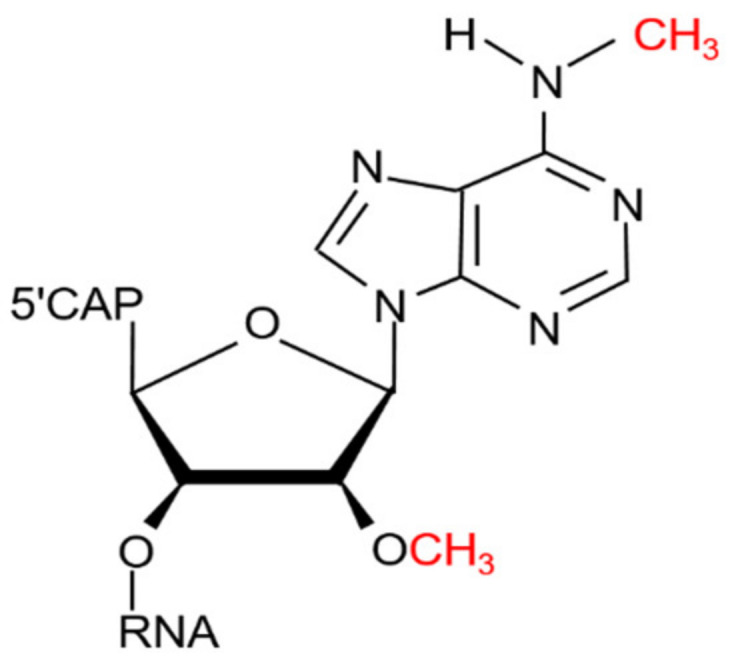
Chemical structures of m^6^Am [18].

**Figure 2 ijms-23-11026-f002:**
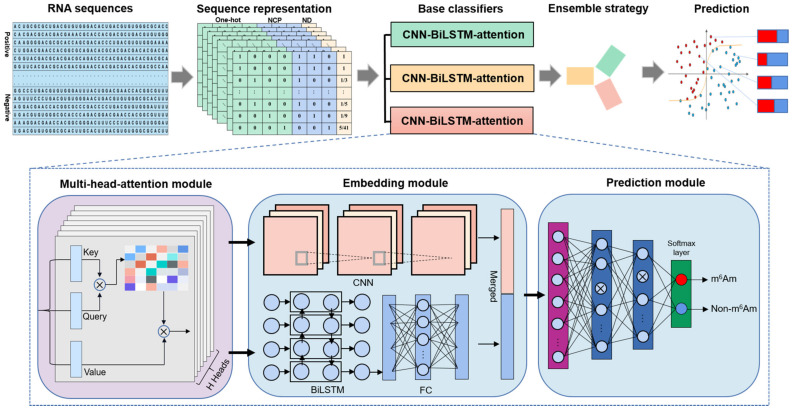
The workflow and architecture of DLm6Am. DLm6Am identifies m^6^Am sites from RNA sequences by several key steps, including feature extraction, model construction, and m^6^Am site prediction. DLm6Am integrates three CNN-BiLSTM-attention models into an ensemble deep learning predictor, in which each base classifier includes multi-head-attention module, embedding module, and prediction module.

**Figure 3 ijms-23-11026-f003:**
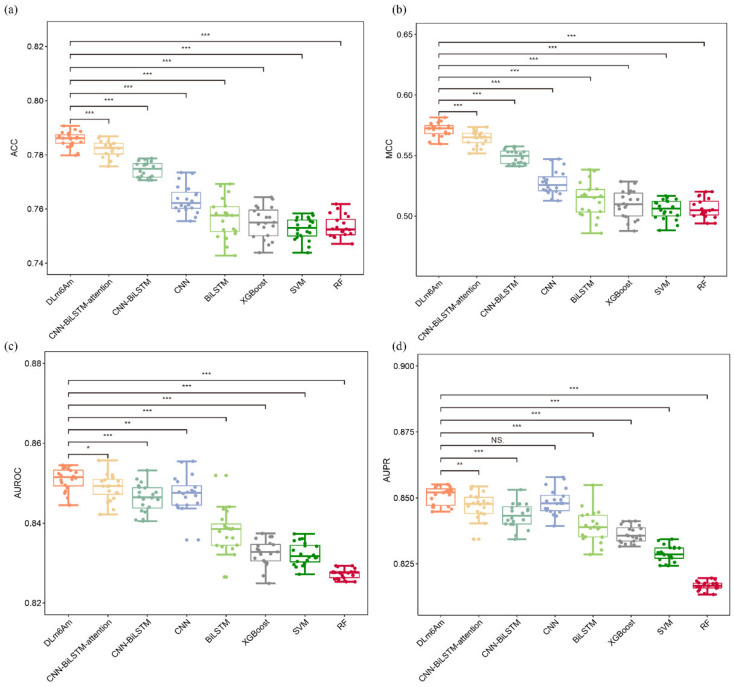
Performance analysis of different m^6^Am prediction models using five-fold cross-validation on training data. Subgraphs (**a**–**d**) represent boxplots of ACC, MCC, AUROC, and AUPR of different models, respectively. The level of significance (NS, *, **, ***) represents non-significant (*p* > 0.05), low significance (*p* < 0.05), medium significance (*p* < 0.01), and high significance (*p* < 0.001), respectively.

**Figure 4 ijms-23-11026-f004:**
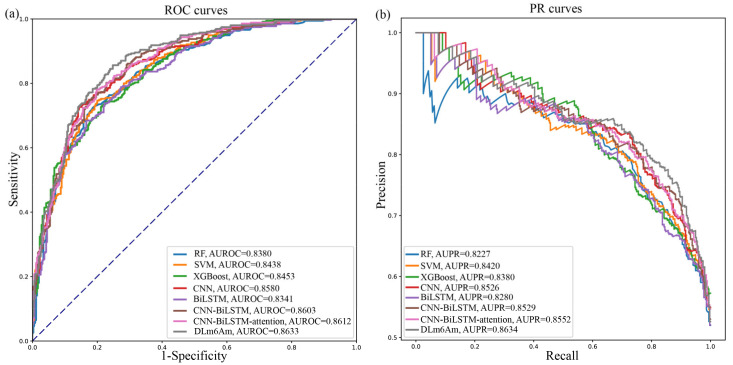
Performance analysis of different m^6^Am prediction models using independent testing data. (**a**) Receiver operating characteristic (ROC) curves of different models. (**b**) Precision–recall curves of different models.

**Figure 5 ijms-23-11026-f005:**
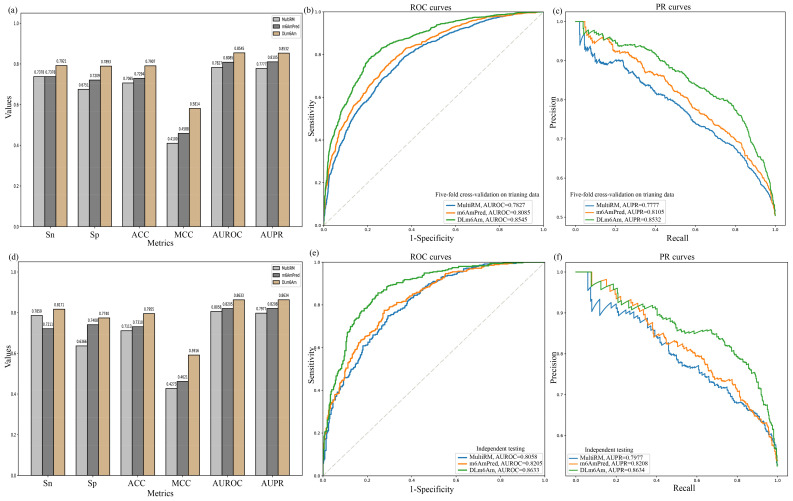
Performance comparison with existing methods. (**a**–**c**) Performance comparison between MultiRM, m6AmPred, and our method DLm6Am using five-fold cross-validation on training data. (**d**–**f**) Performance comparison between MultiRM, m6AmPred, and our method DLm6Am using independent testing data.

**Table 1 ijms-23-11026-t001:** Comparison of performance between different model architectures using five-fold cross-validation on training data.

Model	Sn ± SD (%)	Sp ± SD (%)	ACC ± SD (%)	MCC ± SD	AUROC ± SD	AUPR ± SD
RF	74.74 ± 0.51	75.89 ± 0.54	75.31 ± 0.39	0.5063 ± 0.0078	0.8273 ± 0.0012	0.8163 ± 0.0021
SVM	77.06 ± 0.42	73.46 ± 0.72	75.26 ± 0.39	0.5055 ± 0.0078	0.8323 ± 0.0027	0.8288 ± 0.0029
XGBoost	75.42 ± 0.62	75.54 ± 0.64	75.48 ± 0.55	0.5096 ± 0.0111	0.8325 ± 0.0031	0.8362 ± 0.0031
CNN	75.95 ± 3.21	76.71 ± 2.73	76.33 ± 0.50	0.5275 ± 0.0097	0.8473 ± 0.0043	0.8485 ± 0.0047
BiLSTM	76.17 ± 2.32	75.14 ± 2.06	75.65 ± 0.67	0.5134 ± 0.0134	0.8381 ± 0.0047	0.8392 ± 0.0053
CNN-BiLSTM	78.33 ± 1.08	76.55 ± 1.11	77.45 ± 0.29	0.5491 ± 0.0058	0.8462 ± 0.0034	0.8432 ± 0.0045
CNN-BiLSTM-attention	78.60 ± 0.81	77.82 ± 0.78	78.21 ± 0.29	0.5643 ± 0.0058	0.8488 ± 0.0031	0.8464 ± 0.0053
DLm6Am	78.94 ± 0.79	78.18 ± 0.60	78.56 ± 0.27	0.5713 ± 0.0054	0.8509 ± 0.0027	0.8508 ± 0.0033

**Table 2 ijms-23-11026-t002:** Comparison of performance between different model architectures using independent testing data.

Model	Sn (%)	Sp (%)	ACC (%)	MCC	AUROC
RF	75.77	76.90	76.34	0.5268	0.8380
SVM	75.49	77.18	76.34	0.5268	0.8438
XGBoost	74.37	76.06	75.21	0.5043	0.8435
CNN	69.58	86.20	77.89	0.5656	0.8580
BiLSTM	72.96	78.03	75.49	0.5105	0.8341
CNN-BiLSTM	79.89	77.21	78.55	0.5712	0.8603
CNN-BiLSTM-attention	77.21	80.45	78.84	0.5769	0.8612
DLm6Am	81.71	77.40	79.55	0.5916	0.8634

**Table 3 ijms-23-11026-t003:** The performance of our method using hold-out cross-validation on chromosome level.

Chromosome	Sn (%)	Sp (%)	ACC (%)	MCC	AUROC
Chr1	76.96	80.20	78.61	0.5721	0.8490
Chr2	83.48	76.52	80.00	0.6015	0.8749
Chr3	77.00	79.63	78.37	0.5665	0.8575
Chr4	81.25	78.46	79.84	0.5973	0.8663
Chr5	91.25	81.94	86.84	0.7373	0.9255
Chr6	77.66	76.24	76.92	0.5386	0.8332
Chr7	84.78	77.42	81.08	0.6236	0.8789
Chr8	84.62	80.70	82.57	0.6525	0.8691
Chr9	81.16	80.30	80.74	0.6146	0.8678
Chr10	87.50	77.46	82.52	0.6532	0.8533
Chr11	85.19	75.53	80.69	0.6116	0.8696
Chr12	76.34	82.18	79.38	0.5867	0.8549
Chr14	83.33	77.11	80.00	0.6030	0.8531
Chr15	83.10	86.36	84.67	0.6942	0.9078
Chr16	79.69	83.61	81.60	0.6329	0.8768
Chr17	82.57	77.57	80.09	0.6023	0.8298
Chr19	82.41	77.78	80.19	0.6029	0.8585
ChrX	75.82	90.28	82.21	0.6580	0.9087
Chr13, 18, 20, 21, and 22	87.38	72.36	79.20	0.5979	0.8773

**Table 4 ijms-23-11026-t004:** The distribution of sample numbers in different chromosomes.

Chromosome	Positive	Negative	Chromosome	Positive	Negative
Chr1	191	197	Chr13	32	39
Chr2	115	115	Chr14	72	83
Chr3	100	108	Chr15	71	66
Chr4	64	65	Chr16	64	61
Chr5	80	72	Chr17	109	107
Chr6	94	101	Chr18	16	18
Chr7	92	93	Chr19	108	99
Chr8	52	57	Chr20	37	44
Chr9	69	66	Chr21	18	22
Chr10	72	71	Chr22	26	23
Chr11	108	94	ChrX	91	72
Chr12	93	101	Total	1774	1774

## Data Availability

The training and independent testing used in this study can be downloaded from http://47.94.248.117/DLm6Am/download (accessed on 15 September 2022).

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
