# Peer review of "DLm6Am: A Deep-Learning-Based Tool for Identifying N6,2′-O-Dimethyladenosine Sites in RNA Sequences"

_ijms, 2022, doi:10.3390/ijms231911026_

Round 1

Reviewer 1 Report

Authors present deep-learning architecture (DLm6Am) for predicting m6Am RNA modification sites and compare it with a number of baselines and other methods.

Major comments:

1. Experimental setup:
The model is trained and prediction are performed based on 41nt RNA subsequences.
It would help to evaluate model and its predictions using the full RNA sequences and identify modification sites within those sequences
(e.g., using sliding-window or similar approaches transcriptome-wide).
As it stands currently, the setting used for training/testing is somewhat artificial and may not be practical for studying RNA sequences and their modifications.

2. Model specification
While overall architecture is schematically described in Fig 2, more details should be given regarding actual model configuration(s) used (layer sizes, depth, number of parameters, etc)
One concern is the relatively small size of training datasets vs model complexity/high number of parameters to be estimated.
It would help to evaluate performance while varying model complexity (e.g., number of layers/nodes), as well varying/increasing training set size (see point 1).
How would this affect results / performance?
Also, has pre-training been considered for initializing the model, esp when using smaller training sets?
On the other hand, as can be seen from Table 2, the results for the proposed DLm6Am model are very similar to the baseline models (e.g., in terms of ROC for CNN,CNN-BiLSTM).
It would help to evaluate the additional performance gains of a more complex DLm6Am model vs the baselines in different settings/model configurations/using more data (see 1; also 2) for comparing the results)

2. Results
Performance metrics are reported without deviation estimates. As many results are somewhat similar it would help to also report observed deviations (e.g., Table 2 with cross-validation results). Are the observed differences significant?
When comparing with other methods / baselines it would help to report the statistical significance of the observed differences as some of the differences are minimal.

2. Data availability:
I could not access the data server authors reference in the paper (http://121.36...).
It would be best if the authors deposit their data (training/test sets, etc) along with their results (predictions) into open access Zenodo or Github data repositories. This would facilitate future studies/tools/benchmarks and enable future method developments.

3. Source code availability:
Author mention 'DLm6Am tool' but not providing their tool code. All the code should be provided in, e.g., a Github repository to enable future uses/comparisons.
Necessary documentation/instructions for use should be also provided.

Other comments:
Resolution used in figures is low, esp in figures with results (Figs 3-5), where all the text is hard to read.

Minor comments:
Please proofread. There are occasional grammatical/language issues and typos. E.g.,
Abstract:
"with regulatory in the control" --> regulatory roles?
"three similar base classifier" --> ... classifiers
Introduction:
160 RNA modifications --> 160 RNA modification types
Moreover, data source ... provide --> data sources ... provide
etc

Reviewer 2 Report

The paper relevance and practical usage correspond with the importance of the topic.

Some references should be included at the beginning of the classification method section (2.3). The references MultiRM [34], DeepM6ASeq [37], MASS [38] may be mentioned in the Introduction section as part of the previous work analysis and source of motivations for this new study.

References in the discussion section may also be mentioned in the Introduction.

The configuration and results of the ablation tests should be clarified and presented in details in section 2 and 3.3, respectively.

The effect of diversity may be considered in the ensemble classification results.

The abstract can be improved with more details in the results.
